evolution/genetics/psychology

schizophrenia, Mendelian randomization, stabilizing selection, cliff-edge fitness, reproductive success

**Author for correspondence:**
Rebecca B. Lawn
e-mail: rebecca.lawn@bristol.ac.uk

# Schizophrenia risk and reproductive success: a Mendelian randomization study

Rebecca B. Lawn[1,2], Hannah M. Sallis[1,2,3], Amy E. Taylor[1,2], Robyn E. Wootton[1,2], George Davey Smith[1,3], Neil M. Davies[1,3], Gibran Hemani[1], Abigail Fraser[1,3], Ian S. Penton-Voak[2] and Marcus R. Munafò[1,2]

[1]MRC Integrative Epidemiology Unit at the University of Bristol, Bristol BS8 2BN, UK
[2]School of Psychological Science, University of Bristol, Bristol BS8 1TU, UK
[3]Department of Population Health Sciences, Bristol Medical School, University of Bristol, Bristol BS8 2PS, UK

RBL, 0000-0002-4353-4155; HMS, 0000-0002-4793-6290

Schizophrenia is a debilitating and heritable mental disorder associated with lower reproductive success. However, the prevalence of schizophrenia is stable over populations and time, resulting in an evolutionary puzzle: how is schizophrenia maintained in the population, given its apparent fitness costs? One possibility is that increased genetic liability for schizophrenia, in the absence of the disorder itself, may confer some reproductive advantage. We assessed the correlation and causal effect of genetic liability for schizophrenia with number of children, age at first birth and number of sexual partners using data from the Psychiatric Genomics Consortium and UK Biobank. Linkage disequilibrium score regression showed little evidence of genetic correlation between genetic liability for schizophrenia and number of children ($r_g = 0.002$, $p = 0.84$), age at first birth ($r_g = -0.007$, $p = 0.45$) or number of sexual partners ($r_g = 0.007$, $p = 0.42$). Mendelian randomization indicated no robust evidence of a causal effect of genetic liability for schizophrenia on number of children (mean difference: 0.003 increase in number of children per doubling in the natural log odds ratio of schizophrenia risk, 95% confidence interval (CI): $-0.003$ to 0.009, $p = 0.39$) or age at first birth ($-0.004$ years lower age at first birth, 95% CI: $-0.043$ to 0.034, $p = 0.82$). We find some evidence of a positive effect of genetic liability for schizophrenia on number of sexual partners (0.165 increase in

the number of sexual partners, 95% CI: 0.117–0.212, $p = 5.30 \times 10^{-10}$). These results suggest that increased genetic liability for schizophrenia does not confer a fitness advantage but does increase mating success.

## 1. Introduction

Schizophrenia is a severe, debilitating mental disorder that is substantially heritable [1]. The prevalence of schizophrenia remains stable over populations and time, and yet is associated with lower reproductive success for those diagnosed [1–5]. This creates an evolutionary puzzle: how is schizophrenia maintained in the population despite apparent negative selection? Multiple theories have been proposed to explain this paradox [3,6,7]. One is mutation–selection balance, which suggests that selection against detrimental variants is counteracted by the continuous occurrence of new mutations [8,9]. Another is that effects over many common genetic variants are individually too weak to be under negative selection [1,8,10].

Another popular theory is that stabilizing selection operates, where the optimum fitness level for a trait is approximately at the mean of the trait and fitness declines along a normal distribution on either side of this optimum [3,6,11,12]. A related hypothesis is that schizophrenia-related traits may demonstrate 'cliff-edge' effects on fitness, so that fitness increases with increased expression of the trait until a threshold, where increased expression then results in a steep decline in fitness for some individuals [1,12]. Some have suggested that this peak occurs at levels of symptoms that could result in a diagnosis of schizophrenia, with a reproductive advantage among healthy individuals with an increased genetic liability for the disorder (in the absence of the disorder itself) compensating for the lower reproductive success of those with the disorder itself [1,4,12–14]. Behaviourally, it is possible that higher genetic liability for schizophrenia may be associated with attractive traits (e.g. creativity) and therefore also with a greater number of children [4,13]. For example, schizotypy, a personality measure of schizophrenia-proneness, has been shown to be associated with creativity, short-term mating interest and mating success [4,13,15], while genetic liability for schizophrenia is associated with the increased risk of unprotected sex [16].

Relatives of people with schizophrenia are assumed to have an intermediate level of genetic liability for the highly heritable disorder [17]. Studies into whether cliff-edge fitness maintains the prevalence of schizophrenia have therefore largely focused on family studies. However, despite extensive research, there is no clear evidence of increased fecundity in relatives of individuals with schizophrenia [2,7,17]. Del Giudice [17] argued that family studies underestimate the reproductive benefits of schizophrenia-proneness in the general population. He highlights that relatives share not only genetic liability for schizophrenia but also their environments, which may hinder fitness and result in apparent negative selection [17]. It is therefore important to investigate a potential reproductive advantage of schizophrenia-proneness in the wider population, rather than relying on family studies alone. Moreover, it is important to investigate causal relationships between schizophrenia risk and reproductive success, rather than relying on observational methods previously used, which do not support strong causal inference due to bias from residual confounding and reverse causation [18]. These family studies also suggest that optimum fitness could occur before the appearance of symptoms that might result in the diagnosis of schizophrenia. The principal measure of fitness is the number of children, used in the present study; however, both earlier age at first birth and increased numbers of sexual partners have previously been used as indicators of reproductive success, particularly in developed populations in which contraception is commonly used to control family size [4,8,19,20]. Earlier age at first birth probably results in a longer reproductive period, whereas the number of sexual partners captures mating success and hence potential reproductive success [4].

Recent developments in genetic epidemiology mean that it is now possible to investigate the effects of genetic liability for schizophrenia in the wider population. A genome-wide association study (GWAS) identified 128 independent genetic variants from 108 loci associated with schizophrenia that explained approximately 3.4% of the observed variation in schizophrenia risk [21]. These variants have been used to show that genetic liability for schizophrenia (using a risk score comprising these individual variants) is positively associated with creativity [22]. Evidence for associations between genetic liability for schizophrenia and age at first birth is mixed. Higher genetic liability for schizophrenia was found for those with a young (e.g. below 20 years) age at first birth compared to those with an intermediate age at first birth [23,24]. Another study found no clear evidence for linear or quadratic

associations between a genetic liability for schizophrenia and age at first birth [8]. Two previous studies also used schizophrenia-associated variants to investigate whether genetic liability for schizophrenia is associated with number of children, but results were again inconclusive, perhaps due to limited power [8,25]. The studies showed estimates in the direction of a reproductive advantage, but confidence intervals are typically wide and consistent with no effect [8,25]. Nevertheless, these studies demonstrate how genetic liability for schizophrenia can be measured in the wider population.

We applied a range of methods with roots in genetic epidemiology to test part of the cliff-edge hypothesis. For our main analysis, we examine whether increasing genetic liability for schizophrenia increases reproductive success in a largely post-reproductive population-based sample which is not selected on schizophrenia status, and, therefore, includes very few cases. This linear increase is predicted for part of cliff-edge fitness where a reproductive advantage among healthy individuals with higher genetic liability for the disorder compensates for lower reproductive success of those with the disorder itself. Additionally, given suggestions from family studies that there may be a fitness decline of healthy individuals with high genetic liability for the disorder, we conducted sensitivity analyses to investigate a possible nonlinear relationship where at very high levels of genetic score for schizophrenia liability, there is decreased reproductive success in the absence of schizophrenia itself [2,7,17].

For our main analysis, we calculated genetic correlations using linkage disequilibrium (LD) score regression between genetic liability for schizophrenia and reproductive success, measured as number of children, age at first birth and number of sexual partners. Furthermore, we used genetic variants associated with schizophrenia within a Mendelian randomization (MR) framework to estimate the causal effect of genetic liability for schizophrenia on these measures of reproductive success. MR uses single-nucleotide polymorphisms (SNPs), which are assigned at conception and are mostly independent from other variants or environments. MR therefore overcomes some limitations of observational studies previously used to investigate this evolutionary paradox, by reducing bias from confounding and reverse causation [18]. Finally, we estimated the effect of genetically predicted educational attainment on the number of children and age at first birth. Higher genetically predicted education is known to be associated with fewer children and delayed age at first birth [25–29]. As the present study applied MR in a novel context to this evolutionary paradox, we therefore included educational attainment as an exposure with these two outcomes (using the same outcome datasets used for our schizophrenia analysis) as a positive control.

# 2. Methods

## 2.1. Exposure data

We used independent SNPs associated with schizophrenia ($p < 5 \times 10^{-8}$) from the Psychiatric Genomics Consortium GWAS ($N = 35\,123$ cases and $109\,657$ controls) [21]. The 128 SNPs identified explained approximately 3.4% of the observed variance in schizophrenia risk. A total of 101 SNPs remained due to availability in UK Biobank, availability of proxies and meeting exclusion criteria (see electronic supplementary material). Odds ratios (ORs) and standard errors (s.e.) for the 101 SNP and schizophrenia associations were recorded using GWAS data for Europeans only [30]. The final 101 SNPs and effect estimates for schizophrenia genetic variants are listed in electronic supplementary material, table S1.

For educational attainment, we used SNPs associated with educational attainment ($p < 5 \times 10^{-8}$) from a recent GWAS by the Social Science Genetic Association Consortium [31]. As the GWAS conducted a replication in UK Biobank, effect estimates from the pooled sex analysis of the discovery sample were used to avoid sample overlap. Sixty-seven SNPs were available in UK Biobank data and met exclusion criteria. Effect estimates used for educational attainment genetic variants are listed in electronic supplementary material, table S2.

## 2.2. Outcome data

The exposure-associated SNPs described above were extracted from UK Biobank to derive SNP-outcome associations for our outcome data. Extraction was done using PLINK (v. 2.00) and best guess algorithms for determining alleles (full genotyping information below).

## 2.2.1. Sample

UK Biobank is a population-based health research resource consisting of approximately 500 000 people, aged between 38 and 73 years, who were recruited between the years 2006 and 2010 from across the UK [32]. Particularly focused on identifying determinants of human diseases in middle-aged and older individuals, participants provided a range of information (such as demographics, health status, lifestyle measures, cognitive testing, personality self-report, and physical and mental health measures) via questionnaires and interviews; anthropometric measures, blood pressure readings and samples of blood, urine and saliva were also taken. A full description of the study design, participants and quality control (QC) methods has been published [33].

## 2.2.2. Genotyping information in UK Biobank

The full data release contains the cohort of successfully genotyped samples ($N = 488\,377$). A total of 49 979 individuals were genotyped using the UK BiLEVE array and 438 398 using the UK Biobank axiom array. Pre-imputation QC, phasing and imputation are described elsewhere [34]. In brief, prior to phasing, multiallelic SNPs or those with minor allele frequency (MAF) less than or equal to 1% were removed. Phasing of genotype data was performed using a modified version of the SHAPEIT2 algorithm [35]. Genotype imputation to a reference set combining the UK10 K haplotype and Haplotype Reference Consortium (HRC) reference panels [36] was performed using IMPUTE2 algorithms [37]. The analyses presented here were restricted to autosomal variants within the HRC site list using a graded filtering with varying imputation quality for different allele frequency ranges. Therefore, rarer genetic variants are required to have a higher imputation Info score (Info > 0.3 for MAF > 3%; Info > 0.6 for MAF 1–3%; Info > 0.8 for MAF 0.5–1%; Info > 0.9 for MAF 0.1–0.5%) with MAF and Info scores having been recalculated on an in-house-derived 'European' subset. Individuals with sex-mismatch (derived by comparing genetic sex and reported sex) or individuals with sex-chromosome aneuploidy were excluded from the analysis ($N = 814$). We restricted the sample to individuals of white British ancestry who self-report as 'White British' and who have very similar ancestral backgrounds according to the principal component analysis (PCA) ($N = 409\,703$), as described by Bycroft *et al.* [34]. Estimated kinship coefficients using the KING toolset [38] identified 107 162 pairs of individuals [34]. An in-house algorithm was then applied to this list and preferentially removed the individuals related to the greatest number of other individuals until no related pairs remain. These individuals were excluded ($N = 79\,448$). Additionally, two individuals were removed due to them relating to a very large number (greater than 200) of individuals. QC protocol is described elsewhere [39].

## 2.2.3. Outcome measures

We derived the number of children and age at first birth similarly to previous analyses in UK Biobank [28]. Participants were either asked how many children they had given birth to or how many children they had fathered. We further derived a binary variable to indicate if participants were childless or not (childlessness coded as 1). Age at first birth was only measured in females in UK Biobank, with participants asked: 'How old were you when you had your first child?'. If participants indicated that they had had sexual intercourse, they were asked 'About how many sexual partners have you had in your lifetime?'. Participants were given the information that 'Sexual intercourse includes vaginal, oral or anal intercourse' if they activated the help button. We coded responses to missing if above the 99th percentile. We then derived a binary measure indicating if participants were in approximately the top 10th percentile for the highest number of sexual partners (equal to or above 12 partners coded as 1). Although no age restrictions were applied in analyses, the nature of UK Biobank data meant that participants were aged towards the end of their reproductive lives.

## 2.3. Data analysis

We used LD score regression [40,41] to calculate the genome-wide genetic correlation ($r_g$) between schizophrenia liability or predicted educational attainment and number of children, age at first birth and number of sexual partners. Genome-wide associations were conducted for these outcomes using linear regression, implemented in PLINK v. 2.00 through the Medical Research Council Integrative Epidemiology Unit GWAS pipeline [42]. In this, we adjusted for the top 10 principal components. For the

number of children and number of sexual partners analysis, age and sex were also included as covariates. We then filtered results on MAF (greater than 0.01) and imputation quality (greater than 0.8) separately.

In MR analyses, data were harmonized to ensure that the effect of the SNP on the exposure and the SNP on the outcome corresponded to the same allele. The increasing allele for schizophrenia liability and educational attainment was used. Associations for exposure SNPs and all outcome measures were then calculated in R, fitting the same covariates as listed above. Effect sizes for number of children, age at first birth and number of sexual partners analysis are listed in electronic supplementary material, tables S1 and S2. SNP-exposure and SNP-outcome data were combined using an inverse variance weighted (IVW) approach which is analogous to a weighted regression of SNP-outcome coefficients on SNP-exposure coefficients with the intercept constrained to zero [43], and further assumes all variants are valid instruments or allows pleiotropy to be balanced across instruments when using the random effect [44] with Cochran's Q providing a measure of any overdispersion (see electronic supplementary material).

The IVW effect estimate will only be consistent if all genetic variants in the analysis are valid. Weighted median, mode-based estimator and MR-Egger regression are complementary approaches that can be used to investigate the impact of invalid instruments on our effect estimates. The weighted median estimates a consistent effect estimate if at least 50% of the instruments are valid [45]. The mode-based estimator provides a consistent effect estimate when the largest number of similar individual-instrument estimates come from valid instruments, even if the majority are invalid [46]. A tuning parameter of 0.5 was set for mode-based estimator analysis. One of the main assumptions underpinning MR is that of no horizontal pleiotropy (i.e. no direct effect of the genetic variant on the outcome that does not act through the exposure) [47]. MR-Egger regression analysis can be used to further investigate this; MR-Egger does not constrain the intercept to zero and the intercept term therefore estimates overall directional pleiotropy [48]. We calculated F-statistics (mean of the squared SNP-exposure association divided by the squared s.e. for SNP-outcome association) to indicate the strength of instrument, and $I^2_{GX}$ statistics to assess the suitability of MR-Egger (above 0.9 is desired) [47]. Analysis was repeated after removing the few schizophrenia cases in our sample. All analysis was also conducted with SNP-outcome associations additionally adjusted for genotype array.

MR results were multiplied by 0.693 to represent the causal estimate per doubling in odds of schizophrenia risk [49]. For childlessness and highest number of sexual partners as outcomes, all MR results were multiplied by 0.693 on the log-odds scale, and then exponentiated. The reported estimates therefore indicate the effect of doubling the odds of schizophrenia on the odds of childlessness or being in the highest number of sexual partners category. The effects of education on childlessness were converted to ORs by exponentiating log ORs.

As an illustration of shape of the schizophrenia liability–reproductive success relationship, we created an additive unweighted genetic score for schizophrenia liability in UK Biobank. The score was created in R (v. 3.2.0), with missing SNP data replaced with the mean value for that SNP across individuals. We then divided this score into quintiles and deciles and plotted the mean number of children, age at first birth and number of sexual partners across these categories of the genetic score. We additionally plotted these relationships with reproductive success on the x-axis. As sensitivity analysis to assess if there was any decline in reproductive success within our sample at very high levels of genetic liability, we conducted a series of linear regressions in Stata (v. MP 15.1) between this genetic score for schizophrenia liability and our outcomes, systematically removing cumulative centiles from the maximum. This analysis included adjustment for the top 10 principal components and was repeated after removing the few schizophrenia cases in our sample. Similarly, to further investigate a possible peak in reproductive success at high genetic liability for schizophrenia, we conducted quadratic regression analysis of the genetic score for schizophrenia liability and our outcomes (adjusted for the top 10 principal components and additionally adjusted for sex and age at assessment where appropriate). Lastly, we repeated quadratic analysis separately for each sex.

Analysis scripts are available on GitHub [50].

## 3. Results

In our sample, from UK Biobank, there were more females than males, a majority had children and a minority had college or university degree qualifications (electronic supplementary material, table S3). The mean age was 56.9 years (s.d.: 8.0) and the mean years of education was 13.3 (s.d.: 4.4). For our outcomes, the mean number of children was 1.8 (s.d.: 1.2), mean number of sexual partners was 5.8 (s.d.: 8.6) and the mean age at first birth was 25.4 years (s.d.: 4.5).

## 3.1. Genetically predicted educational attainment

We found a modest negative genetic correlation between educational attainment-associated variants and number of children ($r_g = -0.35$, $p = 8.57 \times 10^{-41}$) and a strong positive genetic correlation between educational attainment-associated variants and age at first birth ($r_g = 0.81$, $p < 5 \times 10^{-41}$) (table 1). The number of individuals in this analysis was 283 723 for educational attainment, 333 628 for number of children and 123 310 for age at first birth data. There was a total of 1 117 154 SNPs included in analysis.

Educational attainment variants showed a mean $F$-statistic (strength of instrument) of 33.23, with above 10 indicating acceptable levels of relative bias (less than 10%) [44,51]. We applied multiple MR methods with IVW results reported throughout the text, and other methods only when not consistent. We found that educational attainment had a negative effect on number of children (mean difference: $-0.16$, 95% confidence interval (CI): $-0.21$ to $-0.12$, $p = 3.63 \times 10^{-10}$ per year increase in educational attainment) and a positive effect on age at first birth (mean difference: 2.68, 95% CI: 2.40–2.95, $p < 5 \times 10^{-14}$) per year increase in educational attainment (table 2). We also found an effect of increased education on increased likelihood of being childless (OR: 1.60, 95% CI: 1.47–1.75, $p = 1.60 \times 10^{-14}$ per year increase in educational attainment). Results for all educational attainment analysis with genotype array included as a covariate in our outcome summary statistics are presented in electronic supplementary material, tables S4 and S5.

## 3.2. Genetic liability for schizophrenia

Using LD score regression, we found little evidence of genetic correlations ($r_g$) between schizophrenia liability and number of children ($r_g = 0.002$, $p = 0.84$), age at first birth ($r_g = -0.007$, $p = 0.45$) and number of sexual partners ($r_g = 0.007$, $p = 0.42$) (table 1). The number of individuals included was 35 123 cases and 109 657 controls for schizophrenia liability, 333 628 for number of children, 123 310 for age at first birth and 273 970 for number of sexual partners data. The analysis included 1 114 456 SNPs.

The mean $F$-statistic for schizophrenia genetic liability was 35.15. There was little evidence that higher genetic liability for schizophrenia increased number of children (mean difference: 0.003 increase in the number of children per doubling in the natural log OR of schizophrenia liability, 95% CI: $-0.003$ to 0.009, $p = 0.39$) or decreased age at first birth ($-0.004$ years lower age at first birth, 95% CI: $-0.043$ to 0.034, $p = 0.82$) (table 2). We further tested childlessness as an outcome and found no strong evidence of an effect of genetic liability for schizophrenia on childlessness (table 2). We found that higher genetic liability for schizophrenia had a positive effect on number of sexual partners (0.165 increase in number of sexual partners, 95% CI: 0.117–0.212, $p = 5.30 \times 10^{-10}$) (table 2). A positive effect was also seen in the analysis of our binary measure for the highest number of sexual partners (table 2). We repeated the MR analysis after removing the few schizophrenia cases in our sample (maximum $N = 207$) with no clear change in results. Results for these analyses with genotype array included as a covariate in our outcome summary statistics are presented in electronic supplementary material, tables S4 and S5.

Our sensitivity analysis investigating a possible nonlinear relationship is presented in figures 1–3, showing the mean number of children, sexual partners and age at first birth for quintiles of an unweighted additive genetic score for schizophrenia liability. Although these figures are somewhat suggestive of a nonlinear relationship between the genetic score for schizophrenia liability and mean age at first birth, there is little evidence of heterogeneity across values of the schizophrenia score. The relationship between the genetic score for schizophrenia liability and number of sexual partners appears more linear. Similar patterns are seen across deciles of the genetic score for schizophrenia liability and when plotting these measures of reproductive success on the $x$-axis (electronic supplementary material, figures S1–S6). A series of regressions between the genetic score and age at first birth, systematically removing cumulative centiles from the maximum, suggests that the relationship is strongest at intermediate levels. It appeared that estimates became slightly stronger in the analysis with number of children, although there was little statistical support (table 3). This analysis was repeated after removing the few schizophrenia cases in UK Biobank (maximum $N = 207$), which did not alter these results (electronic supplementary material, table S6). Regression of the genetic score for schizophrenia liability and number of children showed no clear evidence, also when including a quadratic term for genetic liability for schizophrenia and when stratified by sex (table 4). This quadratic relationship suggested a slight peak in fitness at intermediate levels of the genetic liability, particularly for females, but again with little statistical support (electronic supplementary

**Table 1.** Genetic correlations of genetic liability for schizophrenia and genetically predicted educational attainment on number of children, age at first birth and number of sexual partners using LD score regression.

| | no. of children[a] | | | age at first birth[b] | | | no. of sexual partners[c] | | |
| --- | --- | --- | --- | --- | --- | --- | --- | --- | --- |
| | $r_g$ | s.e. | p-value | $r_g$ | s.e. | p-value | $r_g$ | s.e. | p-value |
| genetic liability for schizophrenia[d] | 0.002 | 0.008 | 0.84 | −0.007 | 0.009 | 0.45 | 0.007 | 0.009 | 0.42 |
| genetically predicted educational attainment[e] | −0.347 | 0.026 | $8.57 \times 10^{-41}$ | 0.805 | 0.019 | $<5 \times 10^{-41}$ | — | — | — |

[a]Number of children data from UK Biobank (N = 333 628).

[b]Age at first birth data from UK Biobank (N = 123 310).

[c]Number of sexual partners data from UK Biobank (N = 273 970).

[d]Schizophrenia data from the Psychiatric Genomics Consortium GWAS (N = 35 123 cases and 109 657 controls).

[e]Educational attainment from the Social Science Genetic Association Consortium GWAS (N = 283 723). There were 1 114 456 SNPs included in schizophrenia analyses and 1 117 154 included in educational attainment analyses.

**Table 2.** Estimates of the causal effect of genetic liability for schizophrenia and genetically predicted educational attainment on number of children, age at first birth and number of sexual partners using IVW, mode-based estimator, MR-Egger and weighted median MR approaches.

| method | β (95% CI), p-value | | | OR (95% CI), p-value | |
| --- | --- | --- | --- | --- | --- |
| | no. of children[b] | age at first birth[c] | no. of sexual partners[d] | childlessness[e] | highest number of sexual partners[f] |
| genetic liability for schizophrenia: 101 SNPs[a] | | | | | |
| inverse variance weighted | 0.003 (−0.003, 0.009), 0.39 | −0.004 (−0.043, 0.034), 0.82 | 0.165 (0.117, 0.212), $5.30 \times 10^{-10}$ | 0.998 (0.985, 1.012), 0.79 | 1.057 (1.038, 1.077), $4.52 \times 10^{-8}$ |
| MR-Egger intercept | −0.001 (−0.004, 0.001), 0.29 | −0.016 (−0.031, −0.001), 0.04 | −0.005 (−0.023, 0.014), 0.61 | 0.998 (0.993, 1.004), 0.55 | 0.994 (0.987, 1.001), 0.08 |
| MR-Egger slope | 0.020 (−0.013, 0.053), 0.23 | 0.214 (0.007, 0.420), 0.04 | 0.229 (−0.024, 0.482), 0.08 | 1.019 (0.950, 1.094), 0.59 | 1.151 (1.043, 1.271), 0.01 |
| weighted median | 0.006 (−0.004, 0.015), 0.23 | 0.023 (−0.042, 0.089), 0.49 | 0.172 (0.092, 0.230), $7.50 \times 10^{-2}$ | 0.995 (0.975, 1.016), 0.65 | 1.034 (1.003, 1.066), 0.03 |
| simple mode-based estimator | 0.020 (−0.014, 0.055), 0.25 | 0.067 (−0.196, 0.331), 0.62 | 0.376 (−0.072, 0.823), 0.10 | 0.988 (0.912, 1.070), 0.76 | 1.121 (0.977, 1.287), 0.11 |
| weighted mode-based estimator | 0.020 (−0.012, 0.052), 0.22 | 0.060 (−0.175, 0.294), 0.62 | 0.389 (−0.032, 0.810), 0.07 | 0.992 (0.924, 1.065), 0.83 | 1.010 (0.884, 1.154), 0.88 |
| genetically predicted educational attainment: 67 SNPs[g] | | | | | |
| inverse variance weighted | −0.162 (−0.206, −0.118), $3.63 \times 10^{-10}$ | 2.677 (2.401, 2.952), $<5 \times 10^{-14}$ | — | 1.589 (1.446, 1.746), $1.60 \times 10^{-14}$ | — |
| MR-Egger intercept | 0.004 (0.001, 0.008), $2.50 \times 10^{-02}$ | −0.031 (−0.054, −0.008), $9.52 \times 10^{-03}$ | — | 0.990 (0.982, 0.997), 0.010 | — |
| MR-Egger slope | −0.391 (−0.595, −0.187), $2.99 \times 10^{-04}$ | 4.348 (3.069, 5.628), $4.12 \times 10^{-09}$ | — | 2.812 (1.813, 4.362), $1.38 \times 10^{-05}$ | — |

(Continued.)

**Table 2.** (Continued.)

| method | β (95% CI), p-value | | | OR (95% CI), p-value | |
| --- | --- | --- | --- | --- | --- |
| | no. of children[b] | age at first birth[c] | no. of sexual partners[d] | childlessness[e] | highest number of sexual partners[f] |
| weighted median | $-0.206$ ($-0.276$, $-0.135$), $2.93 \times 10^{-07}$ | $2.828$ ($2.387$, $3.270$), $<5 \times 10^{-14}$ | — | $1.567$ ($1.343$, $1.829$), $3.00 \times 10^{-07}$ | — |
| simple mode-based estimator | $-0.253$ ($-0.5107$, $0.005$), $0.06$ | $3.454$ ($1.938$, $4.969$), $3.18 \times 10^{-05}$ | — | $1.474$ ($0.884$, $2.457$), $0.14$ | — |
| weighted mode-based estimator | $-0.249$ ($-0.478$, $-0.020$), $0.04$ | $1.649$ ($0.303$, $2.995$), $1.92 \times 10^{-02}$ | — | $1.513$ ($0.952$, $2.404$), $0.085$ | — |

[a]Schizophrenia genetic data from the Psychiatric Genomics Consortium GWAS ($N = 35\,123$ cases and $109\,657$ controls).

[b]Number of children data from UK Biobank ($N = 318\,921 - 335\,758$ for genetic liability of schizophrenia analysis and $268\,658 - 335\,758$ for educational attainment analysis).

[c]Age at first birth data from UK Biobank ($N = 117\,844 - 124\,093$ for genetic liability of schizophrenia analysis and $99\,317 - 124\,093$ for educational attainment analysis).

[d]Number of sexual partners data from UK Biobank ($N = 261\,931 - 275\,700$).

[e]Childlessness data from UK Biobank ($N = 318\,921 - 335\,758$ for genetic liability of schizophrenia analysis and $268\,658 - 335\,758$ for educational attainment analysis). Childlessness was coded as 1.

[f]Highest number of sexual partners data from UK Biobank ($N = 261\,931 - 275\,700$). Highest tenth percentile was coded as 1. Schizophrenia results were multiplied by 0.693 to represent the estimate per doubling in odds of the binary exposure. Results were converted to ORs for schizophrenia by multiplying log ORs by 0.693 and then exponentiating to represent the OR per doubling in odds of the binary exposure. Results were converted to ORs for educational attainment by exponentiating log ORs.

[g]Educational attainment from the Social Science Genetic Association Consortium GWAS ($N = 283\,723$). It should be noted that the $I^2_{GX}$ statistic for an unweighted MR-Egger regression was 0.33 for educational attainment and 0.20 for genetic liability of schizophrenia, which is deemed too low to conduct a SIMEX adjustment, and MR-Egger results should be treated with caution [47].

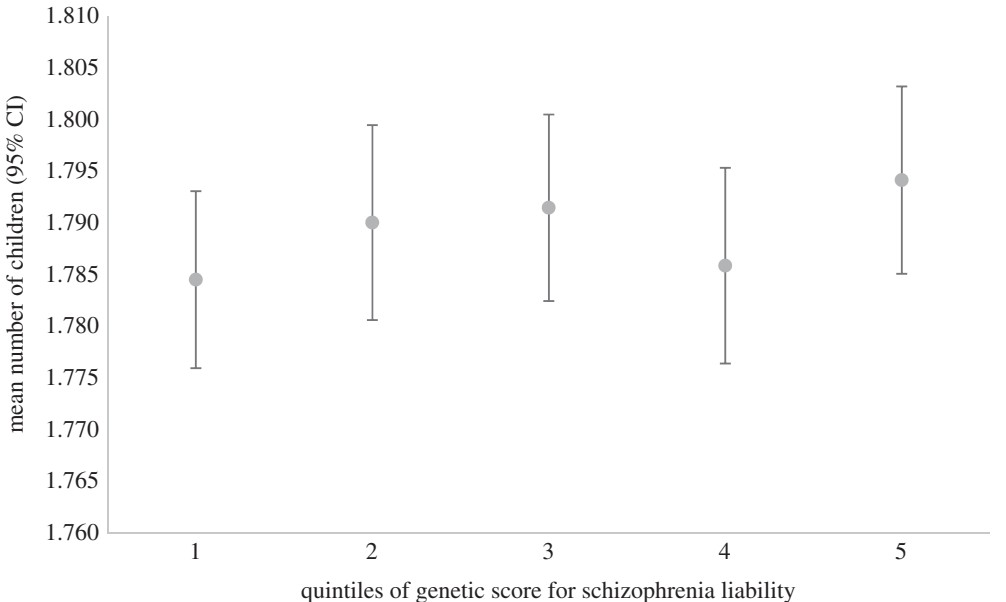

**Figure 1.** Genetic score for schizophrenia liability (in quintiles) and mean number of children in UK Biobank data showing little evidence of heterogeneity across values of the score.

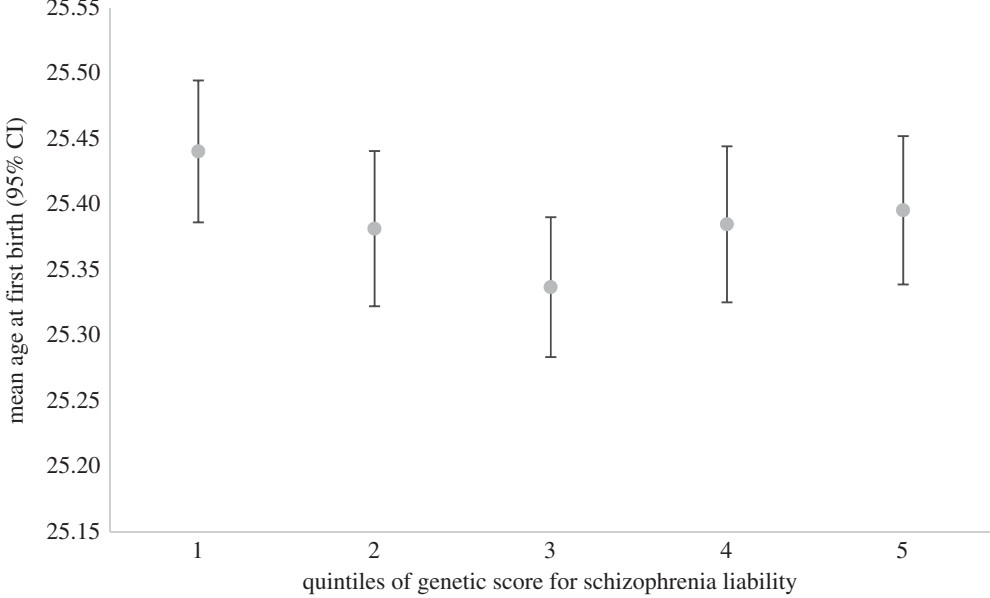

**Figure 2.** Genetic score for schizophrenia liability (in quintiles) and mean age at first birth in women from UK Biobank data also showing little evidence of heterogeneity across values of the score.

material, figures S7–S9). Regression of the genetic score for schizophrenia liability on age at first birth showed no clear evidence; however, there was a weak association when including a quadratic term for the genetic score, suggesting lowest age at first birth was seen at intermediate levels of genetic liability (table 4; electronic supplementary material, figure S10). For the number of sexual partners, the relationship appeared to be linear, with no clear evidence when including a quadratic term (table 4; electronic supplementary material, figures S11–S13)

## 4. Discussion

Our results do not indicate a genetic correlation between genetic liability for schizophrenia and reproductive success using LD score regression, or a linear causal effect on number of children and

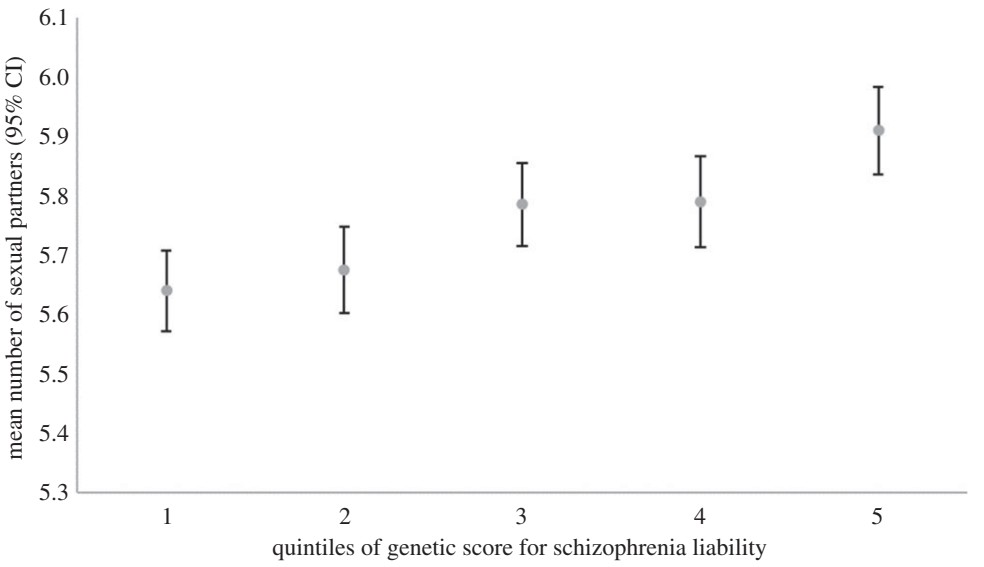

**Figure 3.** Genetic score for schizophrenia liability (in quintiles) and mean number of sexual partners in UK Biobank data suggesting a linear relationship.

age at first birth using MR techniques. This is inconsistent with cliff-edge fitness maintaining schizophrenia in the population, which would predict an increase in fitness with increased genetic liability in the general population. These results support previous research suggesting no strong evidence of a relationship between genetic liability for schizophrenia and number of offspring [8,25]. We find an effect of increasing genetic liability for schizophrenia on increasing number of sexual partners, suggesting liability for the disorder increases mating success in the wider population and could reflect potential reproductive success [4]. Results of our positive control analyses were as expected and in line with previous genetic research, suggesting that educational attainment is under negative selection [25–29], which suggests that the overall approach we adopted here is valid.

Also consistent with previous research, we found no clear evidence for a linear association between genetic liability of schizophrenia and age at first birth [8] and weak evidence of a nonlinear association [24]. In these sensitivity analyses, we found some suggestion of a possible peak in fitness at intermediate to high levels of genetic liability, but there was no statistical evidence for this, suggesting that if this nonlinear association exists, it is very weak, and not reliably detectable even in a large study such as UK Biobank. A previous study also showed little evidence of quadratic associations between genetic liability for schizophrenia and number of children [8]. These sensitivity analyses further suggest that the relationship between genetic liability for schizophrenia and number of sexual partners is linear in this healthy population.

Cliff-edge fitness suggests that schizophrenia prevalence is sustained because the negative reproductive effects in those with an underlying genetic liability and the disorder are offset by a reproductive advantage to those who have an underlying genetic liability but do not develop the disorder [8]. We therefore examined only part of the cliff-edge hypothesis by studying only those without the disorder, testing whether there is a linear effect on reproductive success with increasing genetic liability. Although it is hard to estimate the size of effect on fecundity necessary to sustain the prevalence of schizophrenia (or indeed whether this effect size may fall within the confidence intervals of our estimate), our results provide no support for a cliff-edge fitness effect maintaining schizophrenia prevalence. However, in the modern environment (with available contraception), there are limits to the conclusions we can make about historical evolutionary forces on schizophrenia-associated alleles from these present-day fitness associations. We do find evidence of increased mating success with increasing genetic liability which may proxy reproductive success in past environments and partly explain how the disorder has been maintained in the population. We see stronger evidence for a positive association between genetic liability for schizophrenia and number of sexual partners in males than females, in line with sex differences in reproductive strategies [4]. As variance in reproductive success is higher in males than in females, it has been argued that males obtain higher reproductive benefits from additional matings than females [52]. It has been suggested that creative displays are a form of sexual competition that reflect these evolutionary pressures [4,53]. It is possible that the associations seen in females here are a by-product of male reproductive behaviour, although

**Table 3.** Associations of the genetic score for schizophrenia liability and reproductive outcomes removing cumulative deciles of the score. Adjusted for the top 10 principal components.

| schizophrenia genetic score | number of children | age at first birth | number of sexual partners |
|---|---|---|---|
| | $\beta$ (95% CI), $p$-value | | |
| highest 10% removed | 0.0005 (−0.0003, 0.0013), 0.19 | −0.004 (−0.009, 0.001), 0.15 | 0.011 (0.005, 0.017), $5.57 \times 10^{-4}$ |
| | $N = 302\ 190$ | $N = 111\ 632$ | $N = 248\ 251$ |
| highest 20% removed | 0.0002 (−0.0007, 0.0011), 0.70 | −0.005 (−0.010, 0.001), 0.11 | 0.012 (0.005, 0.020), $9.10 \times 10^{-4}$ |
| | $N = 268\ 604$ | $N = 99\ 134$ | $N = 220\ 847$ |
| highest 30% removed | 0.0005 (−0.0006, 0.0016), 0.35 | −0.008 (−0.014, −0.001), 0.03 | 0.017 (0.008, 0.025), $1.13 \times 10^{-4}$ |
| | $N = 235\ 030$ | $N = 86\ 620$ | $N = 193\ 353$ |
| highest 40% removed | 0.0006 (−0.0006, 0.0019), 0.30 | −0.009 (−0.016, −0.001), 0.02 | 0.015 (0.006, 0.025), $1.98 \times 10^{-3}$ |
| | $N = 208\ 433$ | $N = 76\ 833$ | $N = 171\ 567$ |
| highest 50% removed | 0.0008 (−0.0006, 0.0023), 0.27 | −0.008 (−0.018, 0.001), 0.08 | 0.012 (0.001, 0.024), 0.04 |
| | $N = 167\ 860$ | $N = 61\ 822$ | $N = 138\ 280$ |

**Table 4.** Quadratic regression of the genetic score for schizophrenia liability with number of children, age at first birth and number of sexual partners in UK Biobank data.

| | number of children[a] | age at first birth[b,c] | number of sexual partners[a] |
| --- | --- | --- | --- |
| | $\beta$ (95% CI), p-value | | |
| **genetic score** | | | |
| combined sexes | 0.0002 (−0.0004, 0.0008), 0.53 <br> N = 335 758 | — | 0.016 (0.011, 0.021), $7.6 \times 10^{-11}$ <br> N = 275 700 |
| females | 0.0006 (−0.0002, 0.0014), 0.16 <br> N = 181 255 | −0.001 (−0.005, 0.0027), 0.54 <br> N = 124 093 | 0.005 (0.002, 0.009), 0.004 <br> N = 148 630 |
| males | −0.0003 (−0.0012, 0.0007), 0.60 <br> N = 154 503 | — | 0.029 (0.019, 0.038), $4.9 \times 10^{-9}$ <br> N = 127 070 |
| **including quadratic term for genetic score** | | | |
| combined sexes | 0.0025 (−0.0108, 0.0157), 0.72 <br> N = 335 758 | — | −0.046 (−0.148, 0.056), 0.38 <br> N = 275 700 |
| females | −0.0001 (−0.0174, 0.0172), 0.99 <br> N = 181 255 | −0.088 (−0.171, −0.004), 0.04 <br> N = 124 093 | −0.017 (−0.092, 0.057), 0.65 <br> N = 148 630 |
| males | 0.0055 (−0.0149, 0.0259), 0.60 <br> N = 154 503 | — | −0.085 (−0.289, 0.118), 0.41 <br> N = 127 070 |

[a]Adjusted for the top 10 principal components, age at assessment and sex (in combined sex analysis).
[b]Adjusted for the top 10 principal components.
[c]Age at first birth measured in females only in UK Biobank data.

there are, of course, also benefits to females of attracting additional mates, such as higher mate quality [4]. We assume that, on average, increasing numbers of partners is a reasonable proxy for fitness; however, number of sexual partners has probably also undergone changes since the introduction of contraception, which has allowed for decoupling of sexual and reproductive partners. We therefore cannot conclude that cliff-edge fitness has sustained the prevalence of schizophrenia within the population and provide no evidence for a cliff-edge effect on current fitness.

This leaves us with two further theories for how schizophrenia prevalence is maintained. One is that as schizophrenia is a highly heterogeneous disorder and exhibits a highly polygenic architecture, and effects of genetic variants are individually too weak to be under negative selection [1,8,10]. Our results are consistent with this possibility and suggest that identified schizophrenia risk variants are not under strong selection in the general population. Another explanation is that mutation–selection balance maintains the prevalence of schizophrenia; rare recurrent DNA copy number variants, which are also risk factors for schizophrenia, are filtered out of the population by selection and replenished by de novo mutations [9]. Rare copy number variants conferring risk to psychiatric illness are under strong negative selection [8,9], with most persisting in the population for only two generations [9]. We used results from GWAS, which mainly detect common alleles and therefore cannot determine whether mutation–selection balance sustains the prevalence of schizophrenia through rare variants, although rare variants have been shown to associate with the number of children [1,8]. Other explanations could include an increased likelihood of symptom diagnosis, changes in the environment [54,55] and/or selection bias. UK Biobank data are unrepresentative of the population, given a response rate of approximately 5%, which may introduce selection bias [32,56]. This can generate spurious results in genotypic associations when selection is based on phenotypes associated with the genetic variants and could attenuate associations towards the null if schizophrenia-proneness and increased number of children reduced participation [57–59]. Previous studies have found that higher genetic liability for schizophrenia is associated with lower participation in cohort studies which could bias estimates between genetic liability and traits that lead to non-participation in genetic associations and MR [58,60].

A key strength of this study is the use of MR, which can provide stronger evidence of causality than observational studies [18,61]. We showed agreement between various MR methods that rely on differing assumptions and agreement between methods provides greater confidence in the robustness of the results [62]. We further conducted a positive control analysis to confirm that our approach was valid. Additionally, our MR offers large sample sizes which are necessary for investigating small effect sizes common in such genetic analysis [43]. However, there are also some limitations that should be considered with the current evidence. Firstly, MR relies on genetic variants naturally randomizing an exposure, and therefore inferring causality from genetic liability for schizophrenia as the exposure requires careful interpretation. Our outcome sample was not selected on schizophrenia status, so it contained only few cases of diagnosed schizophrenia. Therefore, we assume that schizophrenia SNPs are associated with subdiagnostic schizophrenia traits that could cause a reproductive advantage within the wider population [4,13,15]. Although debated [63,64], schizophrenia symptoms have been suggested to exist on a continuum, and this assumption could therefore be met [63,65–67]. Within this, we assume that the instrumental variable assumptions are satisfied for this continuous liability to provide a valid test of causality using the binary exposure [49]. Secondly, variants are non-specific, and it is difficult to fully remove population structure, which can induce spurious associations through confounding, even within a sample of European ancestry and adjusting for principal components as we have done [30,68]. Thirdly, age at first birth was measured only in females in UK Biobank, and therefore consists of a different population to the exposure data (which includes data from both females and males). However, the correlation between male and female estimates for age at first birth in a recent GWAS was high [28]. Similarly, the schizophrenia-associated genetic variants used in our sex-stratified analyses were identified from a mixed sex population, although we used an unweighted genetic score to minimize any bias. Lastly, the exposure and outcome samples were each quality controlled for relatedness; however, it is not possible to determine whether participants had relatives across the samples due to the use of summary-level GWAS data for our exposures.

The present study highlights the continued importance of investigating differential fertility and contributes to understanding the maintenance of schizophrenia, and educational attainment, in the population [3,20,69]. Educational attainment has previously been shown to predict human longevity [70] and highlights how even traits with a positive effect on longevity can be maladaptive, although other influences on educational attainment in the population are also identified [29]. This work additionally demonstrates how epidemiological methods can be repurposed to study evolutionary

theories. Future research should investigate causal methods for estimating nonlinear relationships as well as other explanations for this evolutionary paradox, such as mutation–selection balance.

Ethics. UK Biobank received ethics approval from the Research Ethics Committee (REC reference for UK Biobank is 11/NW/0382).

Data accessibility. Genome-wide summary data for schizophrenia and educational attainment can be downloaded from the Psychiatric Genomics Consortium and Social Science Genetic Association Consortium websites (http://pgc.unc.edu; https://www.thessgac.org/). UK Biobank data are available upon application (www.ukbiobank.ac.uk). Analysis scripts are available on GitHub (https://github.com/MRCIEU/Schizophrenia_Fertility_Paper.git).

Authors' contributions. A.F., M.R.M. and I.S.P.-V. conceived the study. R.B.L. conducted the analysis and drafted the initial manuscript. H.M.S., R.E.W., G.H., G.D.S., N.M.D. and A.E.T. assisted with analysis and interpretation. All authors assisted with interpretation, commented on drafts of the manuscript and approved the final version.

Competing interests. The authors declare no competing interests.

Funding. R.B.L., H.M.S., A.E.T., R.E.W., G.D.S., N.M.D., G.H., A.F. and M.R.M. are members of the Medical Research Council Integrative Epidemiology Unit at the University of Bristol, which is supported by the Medical Research Council and the University of Bristol and funds RBL's PhD studentship (grant number: MC_UU_00011/7). A.F. is supported by a personal fellowship from the UK Medical Research Council (MR/M009351/1). A.E.T. and M.R.M. are members of the UK Centre for Tobacco and Alcohol Studies, a UKCRC Public Health Research: Centre of Excellence. Funding from British Heart Foundation, Cancer Research UK, Economic and Social Research Council, Medical Research Council, and the National Institute for Health Research, under the auspices of the UK Clinical Research Collaboration, is gratefully acknowledged. G.H. is funded by the Wellcome Trust (208806/Z/17/Z). The Economics and Social Research Council (ESRC) supports N.M.D. via a Future Research Leaders grant (ES/N000757/1). R.E.W., I.S.P.-V. and M.R.M. are supported by the NIHR Biomedical Research Centre at the University Hospitals Bristol NHS Foundation Trust and the University of Bristol. This research has been conducted using the UK Biobank Resource under Application number 6326.

Acknowledgements. The authors thank Dr Suzi Gage, Dr Jack Bowden, Dr Hannah Jones and Dr Dan Lawson for their technical support and comments. The authors also thank Dr Ruth Mitchell, Dr Gibran Hemani, Mr Tom Dudding and Dr Lavinia Paternoster for conducting the quality control filtering of UK Biobank data. The authors are grateful to the participants of UK Biobank and those who contributed to the PGC and SSGAC GWAS, as well as research staff who worked on the data collection.

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
