## [Reviewer comments · Royal Society Open Science]

Review History

RSOS-181049.R0 (Original submission)

Review form: Reviewer 1

Is the manuscript scientifically sound in its present form?

Yes

Are the interpretations and conclusions justified by the results?

No

Is the language acceptable?

Yes

Is it clear how to access all supporting data?

Yes

Do you have any ethical concerns with this paper?

No

Have you any concerns about statistical analyses in this paper?

No

Recommendation?

Major revision is needed (please make suggestions in comments)

Comments to the Author(s)

Schizophrenia risk and reproductive success: A Mendelian randomization study.
Lawn et al.

This study investigated if increased genetic liability for schizophrenia was associated with reproductive advantage. The authors used GWAS summary statistics from the Psychiatric Genomics Consortium (PGC) for the schizophrenia, which were used in a genetic association analysis (by linkage disequilibrium score regression (LDSC)) and also used in a Mendelian randomization analysis as exposure data. The authors also used UK Biobank data for which reproductive traits (age at first birth and number of children) as well as individual genotyped data were available. The study tested the genetic association and causal effect of schizophrenia genetic risk with number of children and age at first birth and reported that there was no significance.

This is interesting study trying to tackle the old puzzle (how is schizophrenia maintained in the population given its apparent fitness costs?). The authors tested one of plausible reasons that increased genetic risk for schizophrenia is associated with reproductive advantage. However, there is no clear evidence from the analysis results.

I have a number of questions and comments.

1. The authors reported that the genetic correlations from LDSC analysis were not different from zero. Given that the true relationship between schizophrenia (genetic risk) and the reproductive traits, it may not be very good idea to use LDSC which is a linear additive model. For example, the authors should estimate genetic correlation between schizophrenia genetic risk and age at first birth less than its mean value.

A recent study (referenced as #24 in this paper) reported that a significant negative correlation between schizophrenia genetic risk and age at first birth < its mean. They also reported a significant heterogeneity between younger and older age at first birth.

2. The trait, number of children, may be highly (negatively) correlated with age at first birth. Should the authors consider multivariate analysis (age at first birth and # children as multiple dependent variables)? Or, at least did they adjust the effects of age at first birth on # children?

3. How the authors controlled heterogeneity between male and females? I would suggest number of children analysis should be analysed for female only or male only.

4. Figures shows that there is not significant difference of mean # children (or age at first birth) across different schizophrenia genetic liability levels. However, this would be only one dimension of the relationship. The authors should check if there is significant difference of mean schizophrenia genetic liability across different # children or age at first birth.

5. If there are unmeasured confounders, how they will affect the results for the Mendelian randomization analysis? How did the authors control such unmeasured confounders or variables possibly having effects on the exposure and/or outcome data? Did the authors test reverse causal effects, i.e. causal effects of reproductive traits on the schizophrenia genetic liability?

6. I see there was relatedness quality control for UK Biobank (page 5). But how did the authors make sure there was no duplicated or close relatives between PGC and UK Biobank?

7. In Methods, it is not clear how many SNPs were used for the LDSC analyses. How many individuals were used for UK Biobank (for # children and age at first birth analyses)?

Review form: Reviewer 2

Is the manuscript scientifically sound in its present form?

Yes

Are the interpretations and conclusions justified by the results?

Yes

Is the language acceptable?

Yes

Is it clear how to access all supporting data?

Yes

Do you have any ethical concerns with this paper?

No

Have you any concerns about statistical analyses in this paper?

No

Recommendation?

Accept with minor revision (please list in comments)

Comments to the Author(s)

Lawn et al reported in their paper entitled "Schizophrenia risk and reproductive success: A Mendelian randomization study" results of their study investigating whether the stable prevalence of schizophrenia (SCZ) over populations can be explained from some SCZ-induced reproductive advantages. They undertook this task with UK Biobank data for number of children and age at first birth in females and SCZ GWAS summary data using a set of advanced genetic analyses. In summary, no genetic associations were found between SCZ and number of children and age at first birth in females, using both LD score regression and two-sample Mendelian randomization analyses. These results were further observed from the non-significant association between SCZ genetic liability (risk score) and mean number of children and mean age at first birth. They, therefore, suggested two explanations for the stable prevalence of SCZ: SCZ-related genetic variants are too tiny to be under negative selection; and the mutation-selection balance. Overall, this is a very well-written and well-designed and executed study.

One suggestion is to add a caveat in the Discussion that the maintenance about genetic variation

that we observe today (in a contemporary population) is the result of historical evolutionary forces. It is not necessarily the case that the human population is in some kind of equilibrium so that we can infer maintenance of schizophrenia-associated alleles from fitness present-day fitness associations. For example, reproduction has undergone dramatic changes in the last ~50 years because of the availability of contraceptives.

The (minor) specific comments below are provided to improve the manuscript.

1. Page 4 (line 30): The authors utilised education attainment (EA) GWAS published by Okbay et al. in their paper. While to my knowledge, recently there is another much powerful EA GWAS (Lee et al. 2018 Nature Genetics) performed from ~1.1 million individuals. It would be better to re-do the analyses using the latest EA GWAS. However, since the results are unlikely to change substantially the authors should decide whether it is worth the effort.

2. Page 4 ('Exposure data' section): The author should explain more on why also use EA as the exposure data in analyses. For sensitivity comparison? If so, why choose EA, not other psychiatric disorders related to reproduction, such as autism?

3. Page 5 (line 37-38): I wonder if the authors checked the distribution of "number of children" phenotype (i.e. are there any 'outlier' individuals who had a large number of children?). I am also interested in seeing the similarity of "number of children they had given birth to" (female-specific) and "number of children they had fathered" (male-specific) from the true spouses (there are about 1200 known spouse pairs in UK Biobank and about 20,000 inferred spouse pairs), to check the divorce rate which may be a confounder for MR analyses.

4. Page 5 ('Outcome measures' section): One suggestion is to also take reproductive failure (e.g. ever have pregnancy termination, miscarriage) into account in the analyses as a covariate, particularly for the "childless" phenotype.

5. Page 5 (line 41): It is possible to generate a sample for age at first birth in males from identification of potential spouses?

6. Page 5 (line 54-55); Page 6 (line 7-8); Page 7 (line 11): how about fitting all the models (i.e. GWAS, MR, the genetic score-liability regression) with both assessment centre and genotype batch as covariates?

7. Page 7 (line 15): I wonder if it is possible to run MR using "number of children" or "age at first birth" as exposure and EA as outcome, to examine the potential bi-directional association between EA and number of children and age at first birth, depending on whether there is enough independent genome-wide significant ($p < 5e-08$) or suggestive ($p < 1e-05$) SNPs associated with number of children and age at first birth.

8. Table 1 (line 13); Table 2 (line 21 & 24): p-value are not written in a same format.

9. Table 2 (line 14-15): The MR-Egger results for SCZ-age at first birth seems different from other models. The intercept of MR-Egger regression was estimated differently from 0 at a significance level of 5%, indicating the existence of pleiotropic SNP effects. If this is the case, the author should acknowledge it as a potential limitation.

10. Page 8 (line 16): Figure 2 showed some evidence for a 'U' shape curve between SCZ and age at first birth. Since there is a study (Ni et al. 2018 Scientific Reports) published recently also utilised UKB sample and found a 'U' shape relationship between age at first birth in women and SCZ,

maybe it is worth trying to divide the sample into more quantiles (i.e 7 or 9) to check the relationship?

11. Page 11 (line 47): there is a typo in reference #3.

12. Supplementary Figures 1-3 (and description in main text): Comment. This looks like what might be expected under stabilising selection, even if the quadratic term is not statistically significant. Perhaps this could be lack of power? In the future (next few years), the accuracy of schizophrenia liability prediction is going to be better so the issue can be revisited.

Decision letter (RSOS-181049.R0)

02-Nov-2018

Dear Miss Lawn,

The editors assigned to your paper ("Schizophrenia risk and reproductive success: A Mendelian randomization study.") have now received comments from reviewers. While one reviewer is very positive about publication, the other reviewer raises a number of substantive points concerning your analysis which requires careful consideration.

We would like you to revise your paper in accordance with the referees' suggestions which can be found below (not including confidential reports to the Editor). Please note this decision does not guarantee eventual acceptance.

Please submit a copy of your revised paper before 25-Nov-2018. Please note that the revision deadline will expire at 00.00am on this date. If we do not hear from you within this time then it will be assumed that the paper has been withdrawn. In exceptional circumstances, extensions may be possible if agreed with the Editorial Office in advance. We do not allow multiple rounds of revision so we urge you to make every effort to fully address all of the comments at this stage. If deemed necessary by the Editors, your manuscript will be sent back to one or more of the original reviewers for assessment. If the original reviewers are not available, we may invite new reviewers.

- Data accessibility

<http://datadryad.org/submit?journalID=RSOS&manu=RSOS-181049>

- Competing interests

- Authors' contributions

- Acknowledgements

- Funding statement

Please note that Royal Society Open Science charge article processing charges for all new submissions that are accepted for publication. Charges will also apply to papers transferred to Royal Society Open Science from other Royal Society Publishing journals, as well as papers

submitted as part of our collaboration with the Royal Society of Chemistry (<http://rsos.royalsocietypublishing.org/chemistry>). If your manuscript is newly submitted and subsequently accepted for publication, you will be asked to pay the article processing charge, unless you request a waiver and this is approved by Royal Society Publishing. You can find out more about the charges at <http://rsos.royalsocietypublishing.org/page/charges>. Should you have any queries, please contact openscience@royalsociety.org.

on behalf of Dr Steve Brown (Subject Editor)
openscience@royalsociety.org

Comments to Author:

Reviewers' Comments to Author:

Reviewer: 1

Comments to the Author(s)

Schizophrenia risk and reproductive success: A Mendelian randomization study.

Lawn et al.

This study investigated if increased genetic liability for schizophrenia was associated with reproductive advantage. The authors used GWAS summary statistics from the Psychiatric Genomics Consortium (PGC) for the schizophrenia, which were used in a genetic association analysis (by linkage disequilibrium score regression (LDSC)) and also used in a Mendelian randomization analysis as exposure data. The authors also used UK Biobank data for which reproductive traits (age at first birth and number of children) as well as individual genotyped data were available. The study tested the genetic association and causal effect of schizophrenia genetic risk with number of children and age at first birth and reported that there was no significance.

This is an interesting study trying to tackle the old puzzle (how is schizophrenia maintained in the population given its apparent fitness costs?). The authors tested one of plausible reasons that increased genetic risk for schizophrenia is associated with reproductive advantage. However, there is no clear evidence from the analysis results.

I have a number of questions and comments.

1. The authors reported that the genetic correlations from LDSC analysis were not different from zero. Given that the true relationship between schizophrenia (genetic risk) and the reproductive traits, it may not be a very good idea to use LDSC which is a linear additive model. For example, the authors should estimate genetic correlation between schizophrenia genetic risk and age at first birth less than its mean value.

A recent study (referenced as #24 in this paper) reported that a significant negative correlation between schizophrenia genetic risk and age at first birth < its mean. They also reported a significant heterogeneity between younger and older age at first birth.

2. The trait, number of children, may be highly (negatively) correlated with age at first birth. Should the authors consider multivariate analysis (age at first birth and # children as multiple dependent variables)? Or, at least did they adjust the effects of age at first birth on # children?
3. How the authors controlled heterogeneity between male and females? I would suggest number of children analysis should be analysed for female only or male only.
4. Figures shows that there is not significant difference of mean # children (or age at first birth) across different schizophrenia genetic liability levels. However, this would be only one dimension of the relationship. The authors should check if there is significant difference of mean schizophrenia genetic liability across different # children or age at first birth.
5. If there are unmeasured confounders, how they will affect the results for the Mendelian randomization analysis? How did the authors control such unmeasured confounders or variables possibly having effects on the exposure and/or outcome data? Did the authors test reverse causal effects, i.e. causal effects of reproductive traits on the schizophrenia genetic liability?
6. I see there was relatedness quality control for UK Biobank (page 5). But how did the authors make sure there was no duplicated or close relatives between PGC and UK Biobank?
7. In Methods, it is not clear how many SNPs were used for the LDSC analyses. How many individuals were used for UK Biobank (for # children and age at first birth analyses)?

Reviewer: 2

Comments to the Author(s)

Lawn et al reported in their paper entitled "Schizophrenia risk and reproductive success: A Mendelian randomization study" results of their study investigating whether the stable prevalence of schizophrenia (SCZ) over populations can be explained from some SCZ-induced reproductive advantages. They undertook this task with UK Biobank data for number of children and age at first birth in females and SCZ GWAS summary data using a set of advanced genetic analyses. In summary, no genetic associations were found between SCZ and number of children and age at first birth in females, using both LD score regression and two-sample Mendelian randomization analyses. These results were further observed from the non-significant association between SCZ genetic liability (risk score) and mean number of children and mean age at first birth. They, therefore, suggested two explanations for the stable prevalence of SCZ: SCZ-related genetic variants are too tiny to be under negative selection; and the mutation-selection balance. Overall, this is a very well-written and well-designed and executed study.

One suggestion is to add a caveat in the Discussion that the maintenance about genetic variation that we observe today (in a contemporary population) is the result of historical evolutionary forces. It is not necessarily the case that the human population is in some kind of equilibrium so that we can infer maintenance of schizophrenia-associated alleles from fitness present-day fitness associations. For example, reproduction has undergone dramatic changes in the last ~50 years because of the availability of contraceptives.

The (minor) specific comments below are provided to improve the manuscript.

1. Page 4 (line 30): The authors utilised education attainment (EA) GWAS published by Okbay et al. in their paper. While to my knowledge, recently there is another much powerful EA GWAS (Lee et al. 2018 Nature Genetics) performed from ~1.1 million individuals. It would be better to re-do the analyses using the latest EA GWAS. However, since the results are unlikely to change substantially the authors should decide whether it is worth the effort.
2. Page 4 ('Exposure data' section): The author should explain more on why also use EA as the exposure data in analyses. For sensitivity comparison? If so, why choose EA, not other psychiatric disorders related to reproduction, such as autism?
3. Page 5 (line 37-38): I wonder if the authors checked the distribution of "number of children" phenotype (i.e are there any 'outlier' individuals who had a large number of children?). I am also interested in seeing the similarity of "number of children they had given birth to" (female-specific) and "number of children they had fathered" (male-specific) from the true spouses (there are about 1200 known spouse pairs in UK Biobank and about 20,000 inferred spouse pairs), to check the divorce rate which may be a confounder for MR analyses.
4. Page 5 ('Outcome measures' section): One suggestion is to also take reproductive failure (e.g ever have pregnancy termination, miscarriage) into account in the analyses as a covariate, particularly for the "childless" phenotype.
5. Page 5 (line 41): It is possible to generate a sample for age at first birth in males from identification of potential spouses?
6. Page 5 (line 54-55); Page 6 (line 7-8); Page 7 (line 11): how about fitting all the models (i.e. GWAS, MR, the genetic score-liability regression) with both assessment centre and genotype batch as covariates?
7. Page 7 (line 15): I wonder if it is possible to run MR using "number of children" or "age at first birth" as exposure and EA as outcome, to examine the potential bi-directional association between EA and number of children and age at first birth, depending on whether there is enough independent genome-wide significant ($p < 5e-08$) or suggestive ($p < 1e-05$) SNPs associated with number of children and age at first birth.
8. Table 1 (line 13); Table 2 (line 21 & 24): p-value are not written in a same format.
9. Table 2 (line 14-15): The MR-Egger results for SCZ-age at first birth seems different from other models. The intercept of MR-Egger regression was estimated differently from 0 at a significance level of 5%, indicating the existence of pleiotropic SNP effects. If this is the case, the author should acknowledge it as a potential limitation.
10. Page 8 (line 16): Figure 2 showed some evidence for a 'U' shape curve between SCZ and age at first birth. Since there is a study (Ni et al. 2018 Scientific Reports) published recently also utilised UKB sample and found a 'U' shape relationship between age at first birth in women and SCZ, maybe it is worth trying to divide the sample into more quantiles (i.e 7 or 9) to check the relationship?
11. Page 11 (line 47): there is a typo in reference #3.
12. Supplementary Figures 1-3 (and description in main text): Comment. This looks like what

might be expected under stabilising selection, even if the quadratic term is not statistically significant. Perhaps this could be lack of power? In the future (next few years), the accuracy of schizophrenia liability prediction is going to be better so the issue can be revisited.

Author's Response to Decision Letter for (RSOS-181049.R0)

See Appendix A.

RSOS-181049.R1 (Revision)

Review form: Reviewer 1

Is the manuscript scientifically sound in its present form?

Yes

Are the interpretations and conclusions justified by the results?

Yes

Is the language acceptable?

Yes

Is it clear how to access all supporting data?

Yes

Do you have any ethical concerns with this paper?

No

Have you any concerns about statistical analyses in this paper?

No

Recommendation?

Accept with minor revision (please list in comments)

Comments to the Author(s)

The manuscript has been substantially improved and the authors have addressed most of my concerns.

There are some typos, e.g.

Page 17, Line 36.

In References, please check journal name format, e.g. PNAS or Proc Natl Acad Sci? Please also check update some preprints that have been already published in official journals.

Review form: Reviewer 2

Is the manuscript scientifically sound in its present form?

Yes

Are the interpretations and conclusions justified by the results?

Yes

Is the language acceptable?

Yes

Is it clear how to access all supporting data?

Yes

Do you have any ethical concerns with this paper?

No

Have you any concerns about statistical analyses in this paper?

No

Recommendation?

Accept as is

Comments to the Author(s)

The authors have done a good job in their revision. The addition of the number of sexual partners adds interest. Please note that the genetic correlation in liability to schizophrenia between males and females is very high and to my knowledge not significantly different from 1.

Decision letter (RSOS-181049.R1)

17-Dec-2018

Dear Miss Lawn:

On behalf of the Editors, I am pleased to inform you that your Manuscript RSOS-181049.R1 entitled "Schizophrenia risk and reproductive success: A Mendelian randomization study." has been accepted for publication in Royal Society Open Science subject to minor revision in accordance with the referee suggestions. Please find the referees' comments at the end of this email.

The reviewers and Subject Editor have recommended publication, but also suggest some minor revisions to your manuscript. Therefore, I invite you to respond to the comments and revise your manuscript.

- **Ethics statement**

- Data accessibility

If you wish to submit your supporting data or code to Dryad (<http://datadryad.org/>), or modify your current submission to dryad, please use the following link:
<http://datadryad.org/submit?journalID=RSOS&manu=RSOS-181049.R1>

- Competing interests

- Authors' contributions

- Acknowledgements

- Funding statement

Because the schedule for publication is very tight, it is a condition of publication that you submit the revised version of your manuscript before 26-Dec-2018. Please note that the revision deadline will expire at 00.00am on this date. If you do not think you will be able to meet this date please let me know immediately.

Please note that Royal Society Open Science charge article processing charges for all new submissions that are accepted for publication. Charges will also apply to papers transferred to Royal Society Open Science from other Royal Society Publishing journals, as well as papers submitted as part of our collaboration with the Royal Society of Chemistry (<http://rsos.royalsocietypublishing.org/chemistry>). If your manuscript is newly submitted and subsequently accepted for publication, you will be asked to pay the article processing charge, unless you request a waiver and this is approved by Royal Society Publishing. You can find out more about the charges at <http://rsos.royalsocietypublishing.org/page/charges>. Should you have any queries, please contact openscience@royalsociety.org.

Kind regards,
Andrew Dunn

Royal Society Open Science Editorial Office
Royal Society Open Science
openscience@royalsociety.org

on behalf of Prof Steve Brown (Subject Editor)
openscience@royalsociety.org

Reviewer comments to Author:
Reviewer: 2

Comments to the Author(s)

The authors have done a good job in their revision. The addition of the number of sexual partners adds interest. Please note that the genetic correlation in liability to schizophrenia between males and females is very high and to my knowledge not significantly different from 1.

Reviewer: 1

Comments to the Author(s)

The manuscript has been substantially improved and the authors have addressed most of my concerns.

There are some typos, e.g.
Page 17, Line 36.

In References, please check journal name format, e.g. PNAS or Proc Natl Acad Sci? Please also check update some preprints that have been already published in official journals.

Author's Response to Decision Letter for (RSOS-181049.R1)

See Appendix B.

Decision letter (RSOS-181049.R2)

07-Jan-2019

Dear Miss Lawn,

I am pleased to inform you that your manuscript entitled "Schizophrenia risk and reproductive success: A Mendelian randomization study." is now accepted for publication in Royal Society Open Science.

on behalf of Prof Steve Brown (Subject Editor)
openscience@royalsociety.org

Appendix A

Dear Editor,

Thank you for the opportunity to improve our paper by addressing editorial and referees' comments. We have now done so and provide a point by point response to these.

Reviewer 1

1. The authors reported that the genetic correlations from LDSC analysis were not different from zero. Given that the true relationship between schizophrenia (genetic risk) and the reproductive traits, it may not be very good idea to use LDSC which is a linear additive model. For example, the authors should estimate genetic correlation between schizophrenia genetic risk and age at first birth less than its mean value.

A recent study (referenced as #24 in this paper) reported that a significant negative correlation between schizophrenia genetic risk and age at first birth < its mean. They also reported a significant heterogeneity between younger and older age at first birth.

Response: We agree that it is important to consider non-linear approaches here. The cited study aimed to examine whether age at first birth of mothers may predict the schizophrenia risk of offspring, using the interim genetic release of UK Biobank data. In our study, the hypothesis was in the other direction. Here, we aimed to examine genetic liability for schizophrenia on age at first birth where, under a cliff edge-hypothesis, we assume a linear relationship between genetic liability and reproductive advantage in a non-case population (pre-cliff).

As this is an assumption, we also conducted sensitivity analyses to assess non-linear relationships, assessing non-linearity of our independent variable, schizophrenia genetic liability. Previously, we had plotted age at first birth across quintiles of the genetic score for schizophrenia liability. We have now conducted the same sensitivity analysis for age at first birth as done for number of children to examine non-linearity further (Table 4 and Supplementary Table S6; Figure S10). We do see weak evidence for a quadratic association between the genetic score for schizophrenia liability and age at first birth and have now moved this table from supplementary to the manuscript (Table 4).

The cited study also identified an association when not splitting by the mean which we do not find. As the cited study used the interim release of UK Biobank genetic data (N = 38,892), where inclusion in this sub-sample is related to smoking status (Wain et al. 2015, Lancet Respir Med), division is therefore similar to stratifying on smoking status. Individuals with schizophrenia are more likely to smoke and smoking may therefore be a collider and introduce selection bias.

2. The trait, number of children, may be highly (negatively) correlated with age at first birth. Should the authors consider multivariate analysis (age at first birth and # children as multiple dependent variables)? Or, at least did they adjust the effects of age at first birth on # children?

Response: Unfortunately, it is not possible to conduct our Mendelian randomization (our main analysis), with multiple outcomes simultaneously. Nevertheless, we agree that these are likely part of the same causal pathway and are not attempting to understand the effects of number of children that is not via age at first birth.

3. How the authors controlled heterogeneity between male and females? I would suggest number of children analysis should be analysed for female only or male only.

Response: The schizophrenia data are not available by sex and we therefore did not run the no. of children MR analysis stratified by sex, although we did adjust for sex when generating summary

statistics. Although we had also adjusted for sex in quadratic regressions of the genetic risk score to number of children, we have also now added sex stratified regression and quadratic results (Table 4). Although this will suffer from some bias as the genetic variants were identified in a mixed sex population, our use of an unweighted genetic score should be less biased than the weighted MR methods for sex stratified analysis. These regressions test for a causal effect of genetic liability for schizophrenia on number of children and show no clear evidence for either number of births or children fathered. We now highlight the potential limitation of sex-stratified analysis in the discussion. We already present plots in supplementary material that assess the non-linear relationship stratified by sex, again showing little statistical evidence in either sex (Supplementary figures S7-S9).

In Response to reviewer 2, we have now added number of sexual partners as an outcome and include the same sex stratified analysis as for number of children. In this, we find some evidence of a positive effect of genetic liability for schizophrenia on number of sexual partners in combined sexes (mean difference: 0.165 increase in number of sexual partners, 95% CI: 0.117 to 0.212, $p = 5.30 \times 10^{-10}$) (Table 2). We also find a linear positive relationship between genetic liability for schizophrenia and number of sexual partners in males and females separately (Tables 4).

4. Figures shows that there is not significant difference of mean # children (or age at first birth) across different schizophrenia genetic liability levels. However, this would be only one dimension of the relationship. The authors should check if there is significant difference of mean schizophrenia genetic liability across different # children or age at first birth.

Response: As mentioned above, our approach was to examine genetic liability for schizophrenia on age at first birth and number of children, in line with the plots we present in the manuscript. Nevertheless, we have now added these figures to the supplementary material for all main outcomes (Supplementary Figures S4-S6). The figures again show little evidence of heterogeneity across number of children or categories of age at first birth.

5. If there are unmeasured confounders, how they will affect the results for the Mendelian randomization analysis? How did the authors control such unmeasured confounders or variables possibly having effects on the exposure and/or outcome data? Did the authors test reverse causal effects, i.e. causal effects of reproductive traits on the schizophrenia genetic liability?

Response: By using independent genetic variants as instruments, MR exploits Mendel's laws of segregation and independent assortment by which inheritance of genetic variants is determined mostly independently of other genetic variants and the environment. The method therefore reduces bias due to confounding to which non-genetic observational studies are prone. As we used summary GWAS data from the PGC, we could not test this assumption for our exposure data. We did test the genetic score for schizophrenia against environmental factors measured in UK Biobank and observed associations for smoking status and SES (in females). It is possible that these reflect causal pathways (Wootton et al. 2018, doi: <https://doi.org/10.1101/381301>). As our results do not provide clear evidence for the associations we examined, we are less concerned that results are due to confounding, especially in an MR framework. As genotype is determined at conception, MR removes the risk of reverse causality and we therefore did not test reverse effects.

6. I see there was relatedness quality control for UK Biobank (page 5). But how did the authors make sure there was no duplicated or close relatives between PGC and UK Biobank?

Response: Our use of summary level GWAS data from the PGC means that we cannot determine whether participants had relatives across the samples. However, as the PGC uses a case-control

design with samples from several countries we assume the likelihood of having related participants across samples to be low. We now discuss this in the limitations section.

7. In Methods, it is not clear how many SNPs were used for the LDSC analyses. How many individuals were used for UK Biobank (for # children and age at first birth analyses)?

Response: We apologise for not previously including the number of SNPs in LDSC analysis and have now included this in the results section and table caption. We have also included the reported N of individuals from the table caption in the results section. In doing so, we noticed that whilst we were using the estimates from the Schizophrenia GWAS with Europeans only, we had reported the sample size for the GWAS including non-Europeans. We have now corrected this.

Reviewer 2

One suggestion is to add a caveat in the Discussion that the maintenance about genetic variation that we observe today (in a contemporary population) is the result of historical evolutionary forces. It is not necessarily the case that the human population is in some kind of equilibrium so that we can infer maintenance of schizophrenia-associated alleles from fitness present-day fitness associations. For example, reproduction has undergone dramatic changes in the last ~50 years because of the availability of contraceptives.

Response: We agree that there are limits to the conclusions we can make about historical evolutionary forces of schizophrenia-associated alleles from present-day fitness associations, especially with the availability of contraception in evolutionarily recent times. As well as adding text to the discussion on these points, we have also added number of sexual partners as an outcome for analyses. Number of sexual partners has likely also undergone changes since the introduction of contraception, which has allowed for decoupling of sexual and reproductive partners. However, we feel that this outcome adds another dimension to the study of potential reproductive success. Number of sexual partners has previously been used in studies as a measure of mating success and potential reproductive success in modern human populations (reference 4 in the manuscript). These results show a positive effect of genetic liability for schizophrenia on number of sexual partners using MR, although we find no clear evidence of a genetic correlation using LDSC, which has implications for the maintenance of schizophrenia in non-contracepting populations, although any inferences here are speculative (Tables 1 and 2).

The (minor) specific comments below are provided to improve the manuscript.

1. Page 4 (line 30): The authors utilised education attainment (EA) GWAS published by Okbay et al. in their paper. While to my knowledge, recently there is another much powerful EA GWAS (Lee et al. 2018 Nature Genetics) performed from ~1.1 million individuals. It would be better to re-do the analyses using the latest EA GWAS. However, since the results are unlikely to change substantially the authors should decide whether it is worth the effort.

Response: Although the more recent GWAS of educational attainment includes more individuals, it also includes UK Biobank and would therefore cause sample overlap in our analyses. Sample overlap biases the results of MR towards the observed association, therefore we chose to use the earlier GWAS. We also agree that the results with a larger GWAS are unlikely to substantially change our results, given we find robust results using our measure as a positive control.

2. Page 4 ('Exposure data' section): The author should explain more on why also use EA as the exposure data in analyses. For sensitivity comparison? If so, why choose EA, not other psychiatric disorders related to reproduction, such as autism?

Response: As mentioned in the introduction, previous research has robustly shown that increased genetically predicted educational attainment is associated with fewer children and later age at first birth. The present study applied MR in a novel context to test this evolutionary paradox, and we therefore wanted to include a robust positive control with the same outcome dataset used for schizophrenia analysis. Other psychiatric disorders may be the subject of future research, but would not serve as a positive control in this context. We have tried to clarify this in the introduction. Additionally, as there is not robust evidence for an association of genetically predicted educational attainment on number of sexual partners, we have not included this as a positive control.

3. Page 5 (line 37-38): I wonder if the authors checked the distribution of “number of children” phenotype (i.e. are there any ‘outlier’ individuals who had a large number of children?). I am also interested in seeing the similarity of “number of children they had given birth to” (female-specific) and “number of children they had fathered” (male-specific) from the true spouses (there are about 1200 known spouse pairs in UK Biobank and about 20,000 inferred spouse pairs), to check the divorce rate which may be a confounder for MR analyses.

Response: We checked for outliers and although there were participants who reported high numbers of children, these individuals were asked to confirm their answer by UK Biobank (females highest = 22; males highest =100). As number of children, especially for females, is probably an unlikely measure to misreport we kept all individuals in the analysis. We have checked the genetic liability score for schizophrenia in these individuals, and this highest individuals were in the middle range of polygenic risk, therefore unlikely to skew results. We are unsure why the reviewer raises divorce as a concern given that in MR, we use genetic instruments that are unlikely to be related to potential confounders of analyses of observational data.

4. Page 5 (‘Outcome measures’ section): One suggestion is to also take reproductive failure (e.g. ever have pregnancy termination, miscarriage) into account in the analyses as a covariate, particularly for the “childless” phenotype.

Response: It is unfortunately not possible to include a covariate in the MR design used (two-sample MR with summary level data).

5. Page 5 (line 41): It is possible to generate a sample for age at first birth in males from identification of potential spouses?

Response: Although it may be possible to infer at age first birth for some men in UK Biobank, the sample size is likely to be low and likely underpowered to conduct MR analyses. It is also possible that their age at first child does not refer to a child that they had with their spouse in UK Biobank.

6. Page 5 (line 54-55); Page 6 (line 7-8); Page 7 (line 11): how about fitting all the models (i.e. GWAS, MR, the genetic score-liability regression) with both assessment centre and genotype batch as covariates?

Response: We do present LDSC and MR analyses adjusted for genotype batch in supplementary material. We do not predict there to be differences in data collection across centres as we are using self-reported outcomes, and have adjusted for the top 10 principal components to attempt to address population stratification.

7. Page 7 (line 15): I wonder if it is possible to run MR using “number of children” or “age at first birth” as exposure and EA as outcome, to examine the potential bi-directional association between EA and number of children and age at first birth, depending on whether there is enough independent

genome-wide significant ($p < 5e-08$) or suggestive ($p < 1e-05$) SNPs associated with number of children and age at first birth.

Response: Educational attainment is a positive control in this study and not the focus of our analyses. Therefore, although interesting, we do not feel the suggested analyses is within the scope of the current paper.

8. Table 1 (line 13); Table 2 (line 21 & 24): p-value are not written in a same format.

Response: Thank you for highlighting this. The p-values are presented in their current format as they are below what is estimated by the analyses packages used. We therefore decided to present them as less than what we could observe as the minimum for the other measures.

9. Table 2 (line 14-15): The MR-Egger results for SCZ-age at first birth seems different from other models. The intercept of MR-Egger regression was estimated differently from 0 at a significance level of 5%, indicating the existence of pleiotropic SNP effects. If this is the case, the author should acknowledge it as a potential limitation.

Response: Although the results may appear different from other models, we state in the table caption that the I^2_{GX} value is too low for MR-Egger results to be given emphasis here.

10. Page 8 (line 16): Figure 2 showed some evidence for a 'U' shape curve between SCZ and age at first birth. Since there is a study (Ni et al. 2018 Scientific Reports) published recently also utilised UKB sample and found a 'U' shape relationship between age at first birth in women and SCZ, maybe it is worth trying to divide the sample into more quantiles (i.e 7 or 9) to check the relationship?

Response: We have now added figures for all outcomes to supplementary material, using deciles for the genetic score (Supplementary figures S1-S3). As discussed above in response to reviewer 1, we have now included further non-linear analysis for age at first birth to further examine this relationship.

11. Page 11 (line 47): there is a typo in reference #3.

Response: Thank you, we have now corrected this.

12. Supplementary Figures 1-3 (and description in main text): Comment. This looks like what might be expected under stabilising selection, even if the quadratic term is not statistically significant. Perhaps this could be lack of power? In the future (next few years), the accuracy of schizophrenia liability prediction is going to be better so the issue can be revisited.

Response: We agree that these figures appear to show some support for a peak in fitness at intermediate levels of genetic liability for schizophrenia which is consistent with stabilising selection; however, we are reluctant to overstate any findings without clear statistical evidence for number of children. For the present study, we applied relatively novel methods to address this evolutionary paradox, but we look forward to further research as and when these instruments and methods are developed. We thank the reviewer for this comment, highlighting how exciting this research will continue to be in the future.

Appendix B

Dear Editor,

Thank you for accepting our paper, subject to the minor revisions below. We have now addressed the referees' comments and provide responses to these.

Reviewer comments to Author:

Reviewer: 2

Comments to the Author(s)

The authors have done a good job in their revision. The addition of the number of sexual partners adds interest. Please note that the genetic correlation in liability to schizophrenia between males and females is very high and to my knowledge not significantly different from 1.

Response: We thank the reviewer for their comments. We agree that any bias in our sex stratified analysis is likely to be minimal, but we nevertheless highlight the potential limitation in the manuscript.

Reviewer: 1

Comments to the Author(s)

The manuscript has been substantially improved and the authors have addressed most of my concerns.

There are some typos, e.g.
Page 17, Line 36.

In References, please check journal name format, e.g. PNAS or Proc Natl Acad Sci? Please also check update some preprints that have been already published in official journals.

Response: We have now corrected identified typos and errors and updated all preprint references that have since been published in official journals.